# Challenges and Prospects of Sensing Technology for the Promotion of Tele-Physiotherapy: A Narrative Review

**DOI:** 10.3390/s25010016

**Published:** 2024-12-24

**Authors:** Kei Kakegawa, Tadamitsu Matsuda

**Affiliations:** Department of Physical Therapy, Graduate School of Health Science, Juntendo University, Tokyo 113-0033, Japan; fhs8323006@juntendo.ac.jp

**Keywords:** sensing technology, tele-physiotherapy, monitoring, assessment

## Abstract

Significant developments in sensing technology have had many impacts, enhancing monitoring and assessment accuracy across diverse fields. In the field of physical therapy, sensing, which plays a pivotal role in tele-physiotherapy, rapidly expanded amid the COVID-19 pandemic. Its primary objective is to monitor biological signals and patient movements at remote locations. To further enhance the effectiveness and the scope of tele-physiotherapy, it is essential to further develop sensing and data analysis technologies. However, there are usability and analysis issues that have limited its use. The development of these technologies will not only enhance the accuracy of deep learning by AI through the acquisition of big data, but also has the potential to elucidate movement characteristics associated with movement disorders or pathological conditions. Furthermore, improving sensing technologies can broaden applications extending beyond tele-physiotherapy to impact daily life. Looking forward, it holds promise for improving our understanding of disease prognosis and progression.

## 1. Introduction

Motion sensation is useful to comprehend factors influencing change in movement patterns. Its applications extend across various domains, including disease understanding, disability management, preventive measures, and the identification of developmental delay risks. In recent years, sensing technology has undergone significant advancements, expanding its range of applications. The following characteristics are notable in this progress:1.Improvements in sensor devices

Sensor devices have become smaller and lighter, enhancing their usefulness in diverse environments and applications. Improved energy efficiency enables battery-powered devices to be used for longer periods of time.

2.Internet of things (IoT)

Devices are now interconnected, facilitating real-time data collection and analysis [1,2]. Smartphones and healthcare applications allow individuals to monitor their physical activity.

3.Integration with artificial intelligence (AI)

Advanced motion analysis and feedback are now achievable through AI integration.

4.Development of biosensing technology

Biosensing technology has evolved to detect and analyze biological signals, enhancing early detection and response to abnormal signals in daily life, enriching the lives of many. The evolution of the latest sensing technologies has permitted their utilization in various fields, creating new possibilities in the healthcare sector.

The worldwide pandemic of COVID-19 spurred the spread of tele-physiotherapy. Collaboration between engineering and medical disciplines has accelerated the integration of sensing technology in physical therapy, our specialty. While further development of sensing and motion analysis technology is expected to expedite clinical applications in the medical field, challenges might arise, hindering their widespread adoption. To effectively marry technological development with the needs of medicine, it is crucial to understand the current status of sensing and movement analysis technologies, as well as tele-physiotherapy, and address future integration challenges.

As physical therapists, we specialize in observing movement and motion; therefore, focusing on human movement, this paper reviews sensing technologies and feedback, and discusses future issues.

## 2. Purpose

This study aimed to provide a comprehensive review of the literature since 2019, focusing on the development of sensing technologies, motion analysis, and the current understanding of tele-rehabilitation. It also aims to determine the applicability of information obtained from these studies to the field of tele-physiotherapy and identify areas for future research.

## 3. Database

### 3.1. Databases and Search Period

Literature searches were conducted using PubMed and Google Scholar, the major medical and scientific databases. The search period covered the literature from 2019 to 2024 and included English-language literature.

### 3.2. Search Strategy

Searches were performed using the following keywords: “telerehabilitation”, “tele-physiotherapy”, “sensing technology”, and “movement analysis”. Using these keywords, PubMed created a query using the following Boolean operators: (telerehabilitation OR tele-physiotherapy) AND (sensing technology OR movement analysis). In Google Scholar, searches were conducted using multiple-word combinations, and queries were adjusted to include distinct combinations.

### 3.3. Inclusion and Exclusion Criteria

Inclusion Criteria

Studies dealing with sensing technology in tele-physiotherapy or tele-rehabilitation.Articles focusing on movement analysis.Exclusion CriteriaConference proceedings, opinion articles, and book chapters.

### 3.4. Literature Selection Process

Of the 1109 articles obtained from the search results, duplicates were removed and excluded through a review of titles and abstracts. The full text was then reviewed, and finally, 16 references were included in the analysis.

### 3.5. How the Results Were Organized

The selected literature was categorized thematically and reviewed based on research focus, intervention content, and sensing technology.

## 4. Results

### 4.1. Development and Clinical Application of Motion Analysis Technologies

#### 4.1.1. Status of Sensing and Motion Analysis Technology Development

Various sensors, such as light, temperature, acceleration, pressure, magnetic, biological, and acoustic sensors, are employed in sensing technology. These sensors capture real-world information about the environment, objects, and organisms and convert it into digital data [3,4,5], which are applied in not only the medical field but various other fields [6,7,8,9].

A typical example is motion analysis technology’s use in the fields of rehabilitation and ergonomics [10,11]. Three-dimensional motion analysis, employing infrared reflection methods, facilitates precise real-time measurement of body and object movement patterns, providing information that aids in the study of sport biomechanics and evaluation of rehabilitation programs [12,13]. Advancements in sensor technology enable more accurate data acquisition and analysis, contributing to a variety of health-related and engineering applications. Challenges exist, however, such as the high costs of equipment, the time-burden in marker attachment, and the potential displacement or shifting of markers when affixed to skin or clothing. To overcome such challenges, marker-less motion capture technology, which does not require the attachment of reflective markers, is increasingly employed. Marker-less and wireless motion sensors can be used for a wide variety of purposes, from athlete motion analysis in the sport domain to tele-physical therapy in the health domain. Advancements in technologies such as AI and skeletal estimation models, as represented by Openpose, have made simple motion analysis possible.

Studies evaluating technologies like VICON, a 3D motion analyzer, and marker-less motion capture have reported both reliable results [14] and discrepant results [15]. A single camera is prone to errors in depth measurement, and in cases where limb movements overlap with those of the trunk, analysis is difficult and significantly less accurate. Therefore, the measurement of movements includes some limitations. To compensate for some limitations, methods using multiple cameras have been developed, and the efficacy of measuring everyday movements has improved [16]. A comparative study of Openpose and VICON measurements, in which 2D gait data captured by two cameras was converted to 3D data using a gait system, showed a high degree of agreement and was found to be a promising alternative to 3D gait analysis systems [17].

#### 4.1.2. Clinical Application of Movement Analysis in Physical Therapy

In a study conducted by Takeda et al., the accuracy of MAC3D was compared with that of OpenPose and Kinovea software for individuals with impaired lower limb functions crucial for walking. Marker-less motion capture was utilized to assess both normal and abnormal gait patterns, including in those with braces [18].

Similarly, a machine learning application has been developed to measure jump height using marker-less motion capture (OpenPose) [19], as well as to monitor and provide feedback on home physical therapy. This system analyzes movement using posture estimation from two points, a vector and a temporal component, employing inertial sensors with an Android smartwatch. Research indicates this system accurately measures movement and provides reliable feedback to patients and therapists [20].

Gait assessments utilizing smartphones and accelerometers have demonstrated the ability to detect significant differences in gait parameters among healthy adults, facilitating early detection of changes and clinical decision making [21]. This digital gait assessment enhances disease monitoring, such as in multiple sclerosis, and serves as an effective tool for the early identification of changes in a patient’s condition. Additionally, an adaptive empirical pattern transformation application has been used to analyze gait patterns from smartphone videos, showing reasonable accuracy in identifying individuals unable to walk long distances based on gait speed and rhythm assessments [22].

The current status of analysis technology based on motion sensing has been summarized. Validation has extended beyond walking motions to include jumping and various other motions among individuals with various diseases. Further clinical applications are possible with the expansion of motion sensing capabilities and the understanding of motor characteristics associated with different diseases.

### 4.2. Current Status of Tele-Physiotherapy

Telemedicine is an important interface for providing medical care to patients with various diseases. Tele-rehabilitation, which has been employed in clinical settings overseas since the early 21st century, is now emerging as a significant area of research. A database search for articles with “tele-rehabilitation” or “tele-physiotherapy” in the title revealed there were 2205 articles in the last five years alone, indicating a rapid surge in recent clinical studies (Figure 1). This rapid growth in the number of such clinical studies reflects the increasing expectation and interest in tele-physiotherapy, a field expected to continue evolving.

Many recent studies have reported the effectiveness and safety of tele-physiotherapy in various diseases. This section reviews the current status of tele-physiotherapy and use of sensing technology in different disease categories (Table 1).

#### 4.2.1. Neurological Diseases: Tele-Rehabilitation for Chronic and Progressive Conditions

A 12-week tele-rehabilitation program for stroke patients showed significant improvement in Fugl–Meyer Assessment scores at the end of the intervention compared to the conventional rehabilitation group, indicating a beneficial effect of tele-rehabilitation on motor function in stroke patients. Also, the program’s safety was suggested, highlighting its feasibility [23].

Furthermore, tele-rehabilitation is an option for individuals with Parkinson’s dis-ease (PD), a common neurodegenerative condition [38]. Studies have reported tele-rehabilitation to be feasible and effective in the home setting, particularly in improving gait, balance, and upper limb dexterity [25,39,40]. Edoardo Bianchini et al. reported that combining tele-rehabilitation with independent training effectively improved motor symptoms in individuals with mild PD [25].

Many patients with neurological diseases face residual paralysis upon discharge or deal with progressive neurological decline and require ongoing support at home. Accordingly, tele-physiotherapy has an important role to play. While many studies have shown tele-physiotherapy to be effective in maintaining and improving function, few studies have involved sensing, which leaves room for further research.

#### 4.2.2. Cardiovascular and Respiratory Diseases Rehabilitation: Expanding Sensing and Remote Monitoring

Tele-physiotherapy is also used in cardiac rehabilitation support post-discharge from hospital, with the aim to prevent recurrence. In a study using a remote biosignal monitoring system, which enables real-time transmission of various measurement to operators via Bluetooth, deficiencies in remote monitoring were observed. Moreover, significant improvements in exercise capacity were observed, affirming the safety and feasibility of remote cardiac rehabilitation using such a system [29].

Regarding respiratory rehabilitation in patients with chronic obstructive pulmonary disease, the benefits of alternative approaches, such as remote rehabilitation, have been investigated despite challenges such as frequent interruptions. It was reported that more patients completed pulmonary tele-rehabilitation than traditional pulmonary rehabilitation, although there were no group differences in physical function by intervention at the 22-week follow-up [30].

In some cases, COVID-19-infected patients required respiratory physiotherapy, which had a significant impact on the provision of respiratory rehabilitation in tele-physiotherapy. Data such as blood pressure, heart rate, SpO2, and activity level are considered relatively easy to sense and transfer data for. Therefore, sensing technology tends to be utilized more readily for cardiovascular and respiratory diseases than for other diseases.

#### 4.2.3. Orthopedic Diseases: Movement Feedback and Digital Care Pathways

In a study by Sang et al. [9], complete tele-physiotherapy for chronic shoulder pain in musculoskeletal conditions showed significant improvement comparable to face-to-face interventions. Given the scalability, feasibility, and efficacy observed in the study results, there is potential for a digital care pathway to deliver physical therapy, alleviating the current burden of chronic shoulder pain. The study reported fewer interruptions, which may contribute to improved adherence and continuity in the target population.

Knee osteoarthritis, being a progressive disease, often benefits from exercise at home to control symptoms. A study demonstrating the potential to reach a large number of people by using the Internet found that remote exercise guidance from a therapist yielded significant differences and greater improvements in physical function, pain reduction, and satisfaction than self-directed exercise [34].

The latest sensing technology is often employed to analyze movement in athletes, and movement analyzers are used in physical therapy in hospitals. For orthopedic conditions, movement feedback is important for pain control and the prevention of ailment onset and recurrence. However, movement is still not adequately evaluated in tele-physiotherapy. This remote, expert-guided approach enables patients to undergo rehabilitation at home or in remote areas, facilitating improved function and recovery. Furthermore, it is effective not only in cases of limited access to medical care, but also in times of disaster.

### 4.3. Current Status of Sensing in Tele-Physiotherapy

Effective tele-physiotherapy relies on the continuous monitoring of patients to ensure safe and efficient interventions. Various sensors are used in this monitoring process, enabling the evaluation of movement patterns and physiological parameters such as heart rate and blood pressure during exercise load, similarly to physical therapy provided in the hospital. Remote monitoring allows therapists to assess the condition of patients to provide timely guidance and treatment modifications as needed, resulting in quantifiable effective physical therapy. Pairing sensing technology with information communication technology enables bidirectional flow of information in real-time, even remotely [41].

Tele-physiotherapy platforms commonly utilize video call platforms, specialized platforms in medical institutions, and wearable devices to connect patients and professionals. In the reports on tele-rehabilitation reported to date (see references above) [42], video call platforms available to all are often used. However, such tools alone allow therapists to merely watch patients’ movements, making it difficult to assess movements accurately in sufficient detail to provide the most appropriate instructions or modifications. This is where sensor devices can be useful.

The advent of Society 5.0, characterized by the interconnectedness of people and objects through the IoT, where knowledge and information are shared and new products never before seen are created, presents opportunities to address various issues [43]. In particular, these include issues such as a shortage of labor and constraints on activities due to disabilities accompanying age in an aging society; a society with fewer children; depopulation of rural areas; and other issues. The marketplace penetration of IoT wearable sensors and devices and experiments have grown significantly since 2010 [2]. These technologies have found extensive application in tele-physiotherapy, enabling biological monitoring for physical evaluation, safety assessment, and tailored rehabilitation programs based on individual needs [44]. The versatility of these applications is evident in their use for diverse purposes, such as activity recognition, stroke rehabilitation, cardiac and respiratory monitoring, and sleep and stress monitoring, as well as enhancing medical adherence and detecting behavioral abnormalities in Alzheimer’s disease patients [1]. Accumulated data enable the visualization of improvement and change in behavior over time. This growing comprehensive data set, as opposed to data collected from one isolated intervention, allows both the patient and therapist to retrospectively analyze performance, facilitating effective exercise guidance and suggestions based on a holistic view of the patient’s progress.

In a separate study evaluating exercise efficacy at home and other remote locations, researchers investigated the use of wearable inertial sensors. They found that while some degree of error and caution is required in interpreting the results, the convenience of the device makes it suitable for monitoring home exercise programs [45]. However, users should be aware of its accuracy limitations and understand its measurement system before using it for physical therapy recommendations.

Furthermore, integrating biometric data has been proposed to enhance tele-rehabilitation and enable more comprehensive remote physical assessments. There are also reports indicating improved patient adherence and participation, leading to enhanced safety, rehabilitation progress assessments, and outcomes [44].

However, the need for improvement and challenges in applying sensing and movement analysis technologies remains to make tele-physiotherapy more effective. The following sections summarize these challenges and discuss the outlook for the future.

## 5. Current Challenges

The challenges associated with applying sensing and motion analysis technologies to enhance the effectiveness of tele-physiotherapy can be divided into two major areas: usability of sensing devices and analysis and feedback of results.

### 5.1. User Acceptability of Sensing Devices

The primary concern is acceptability, which has been cited as an issue in many studies. Acceptability includes ease of use, comfort, size, and battery life for both hardware and software. In particular, with regard to convenience, many devices, although IoT-enabled, require Bluetooth connection to a PC. There is a high demand for devices that can be easily managed by users, such as those that can be placed at key sites with seamless connectivity. Should this be difficult, we suggest equipment that could be set up and controlled remotely to optimize use of time without delays. Achieving the optimal balance between user comfort and technical performance, including accuracy, is important [2], but it necessitates further improvements in usability.

### 5.2. Handling of Sensed Data: Analysis of Results and Feedback

The next issue is the handling of acquired data. Although the use of sensors is deemed significant in fostering compliance with home exercise, challenges such as the necessity of feedback to regulate movement speed have surfaced [45]. In a study examining movement accuracy using IoT-enabled sensors, it was shown that increasing system accuracy could potentially enhance these sensors’ use in tele-physical therapy. It was suggested that this system might contribute to more tele-physical therapy use [46]. Furthermore, motion analysis validation through marker-less motion capture has typically focused on assessing angles, motion direction, and temporal factors. This analysis has proven useful in detecting change in motion due to disease progression and deviations from normal motion. Feedback is essential in guiding patient goal attainment [26], and accurate assessment is crucial in this. However, assessments pose challenges in inter- and intra-rater reliability. Improved assessment scoring accuracy and accurate information can potentially yield the same effectiveness as face-to-face rehabilitation [47].

For proper evaluation, the establishment of analytical algorithms via improved analytical technique is necessary. This requires clarification of what data and what thresholds are essential to inform and indicate the expected desirable outcomes. Expanding the range of items evaluated could broaden the applicability of the tools. Some studies [46] use dynamic time warping to evaluate motion congruency; however, although medical experts can name the items they wish to evaluate based on their clinical experience, they lack proficiency in utilizing analysis methods to express the results. Conversely, engineering experts may lack insight into the specific evaluations required by medical professionals. Strengthening medical–engineering collaboration and engaging in detailed discussions are essential to align their needs and methodologies. Subsequently, standardizing evaluation tools and scales may reduce barriers to tele-physiotherapy approaches.

## 6. Future Prospects

We anticipate that resolving the discussed issues will broaden the use of sensing technology in tele-physiotherapy. Beyond merely connecting therapists and patients remotely, integrating biological-response-sensing systems into daily life will expand the possibilities for lifestyle support [32]. For example, fluctuations in symptoms such as those in PD drug therapy may be managed more effectively. Distance rehabilitation can contribute to functional improvement, allowing patients to receive online support in their own homes from therapists located elsewhere, overcoming challenges in attending rehabilitation facilities in person [30]. Creating conditions enabling patients to take advantage of this benefit and choose flexible modes of physical therapy will likely drive up the demand for such services. Consequently, the application of sensing and motion analysis technologies will expand, broadening the potential for delivering more effective physical therapy.

Further experiments are needed to construct a biological-response-sensing system in daily life. One of the specific symptoms of Parkinson’s disease, for example, is the inability to move the foot forward in response to intention, which is experienced by about half of the patients and reported to be present in 65% of patients in the advanced stages of the disease [48]. External stimuli such as strides, landmarks, sounds, and vibrations can be helpful [49]. Non-invasive sensing devices may have practical potential for predicting and dealing with this condition. External stimuli such as strides, landmarks, sounds, and vibrations can help in addressing this symptom. By capturing and predicting signs of foot dragging, applying stimuli at optimal times, and encouraging individuals to recognize symptoms, it may be possible to reduce their occurrence and discomfort. Non-invasive sensing devices hold practical potential for predicting and addressing foot dragging.

While AI integrated is already underway, there is significant room for further utilization and improvement in accuracy. Simple sensing systems have enabled the collection of biometric information from a larger population. Leveraging these data could enhance the accuracy of AI-driven deep learning, facilitating a better understanding of disease-specific behaviors, future recovery prospects, and disease progression. By constructing deep learning models, we can refine the accuracy of tailored exercise guidance and support exercise continuation, thereby compensating for areas that therapists cannot manage.

Addressing these diverse challenges requires collaboration among medical and engineering specialists. We believe such collaboration would aid the development of products to meet a wider range of needs, contribute to improving quality of life, and promote their use in society.

## Figures and Tables

**Figure 1 sensors-25-00016-f001:**
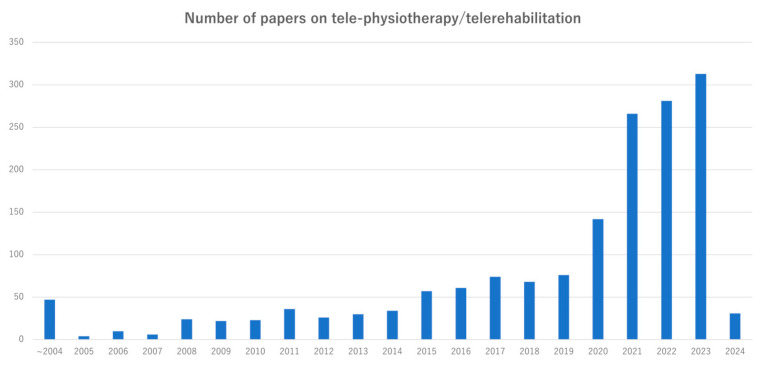
Number of articles with tele-rehabilitation or tele-physiotherapy in the title searched in PubMed.

**Table 1 sensors-25-00016-t001:** Research on the current status of tele-physiotherapy and sensing in various diseases.

**Study and Year**	**Research Design**	**Patients (Sample Size, Intervention, Control)**	**Intervention**	**Main Outcome and Result**	**Connecting/Sensing Devices**
Jing Chen (2020) [23]	Randomized controlled trial	Stroke and hemiplegia (*n* = 26, 26)	10 sessions/week, 60 min of OT and PT and 20 min of ETNS for each session.	Compared to the CR group, the TR group showed significant improvement in FMA at the end of rehabilitation. (*p* = 0.011)	Telemedicine rehabilitation system
Shih, Hai-Jung Steffi (2022) [24]	Single cohort	PD patients with Hoehn and Yahr stage I–III (*n* = 62)	The Engage-PD intervention: up to 5 personal coaching sessions delivered via telehealth, over a 3-month period. A minimum of 3 times/week and a total of 150 min.	recruitment: 62%, retention: 85%. PA Improvement: d = 0.33, ESE: d = 1.20, Goal Performance: d = 1.63, Satisfaction: d = 1.70.	Zoom
Edoardo Bianchini (2022) [25]	Prospective, open-label pilot study	PD patients with Hoehn and Yahr stage < 3 (*n* = 23)	5-week tele-rehabilitation program consisting of a remote session with PT once weekly and at least two self-conducted sessions per week.	Feasibility and safety of tele-rehabilitation—dropout rate: 0%; 85% of patients reached acceptable adherence, 70% optimal adherence; no adverse events reported.	Salute Digitale, audio, video remote conference call interface
Alejandro García-Rudolph (2023) [26]	Matched case–control study	Stroke (*n* = 35, 35)	3.5 h/day rehabilitation (1) TeleNeuroFitness (TNF): 60 min, group exercise; (2) TeleNeuroRehab (TNR): 30 min; (3) TeleNeuroMov (TNM): 15-min exercise videos.	FIM, BI—the groups showed no significant differences in gains, efficiency, or effectiveness using either FIM or BI.	Jitsi Meet, Zoom, Teams
Sefa Eldemir (2023) [27]	Randomized controlled trial	PD patients with Hoehn and Yahr stages I–III (*n* = 15, 15)	In-home training with video sessions and home exercises to improve balance, gait, and mobility 3 days/week for 6 weeks.	Upper extremity motor functions (9-HPT, JHFT, grip strengths, pinch strengths, UPDRS-III): Significant group-by-time interactions were found for the 9-HPT, JHFT, grip strengths (*p* < 0.001), pinch strengths (*p* ≤ 0.015), and UPDRS-III (*p* = 0.007), favoring the TOCT-TR. The TOCT-TR improved upper extremity motor functions, ADL, and QoL in PwPD.	
Elena BEANI (2023) [28]	Randomized controlled trial	Unilateral cerebral palsy (*n* = 15, 15)	Tele-UPCAT (Tele-monitored UPper Limb Children Action Observation Training) platform for home rehabilitation for 1 h per day for 15 consecutive days.	Tele-UPCAT immediate and after effects; AHA: significant improvement in AHA with an effect size of 1.99 (*p* = 0.021), sustained over the medium and long term.	Tele-UPCAT platform/a sensorized toy and wearable sensors
Yokoyama (2023) [29]	Single-center prospective pilot study	Cardiovascular disease (*n* = 9); three of them had undergone open-heart surgery, three had angina pectoris, and three had peripheral arterial disease.	20–40 min of aerobic exercise and body weight resistance training, lasting a total of 3 h/session. For 60–80 min, operators talked with patients via video call, providing information on various symptoms and guidance on disease management, including nutritional guidance.	Lack of remote biological signal monitoring; Peak VO_2_: no issues with ECG waveforms (0%). Significant improvement in Peak VO_2_ (19.5 mL/kg/min → 21.1 mL/kg/min, *p* = 0.01) and anoxic threshold (13.0 mL/kg/min → 15.1 mL/kg/min, *p* < 0.01) after 3 months.	NIPRO Heartline™ connected to blood pressure monitor and heart rate monitor via Bluetooth
Henrik Hansen (2020) [30]	Randomized multicenter trial	COPD (*n* = 49, 42)	3/week for 10 weeks. Pulmonary tele-rehabilitation via a videoconferencing software system; 35 min of exercise and 20 min of patient education/session.	6-min walk distance (6MWD): no differences between groups in change in 6MWD after intervention (9.2 m, 95% CI: −6.6 to 24.9) or at 22 weeks follow-up (−5.3 m, 95% CI: −28.9 to 18.3).	Videoconference software system/single touch screen
Racodon M (2023) [31]	Retrospective research	Ischemic heart disease, acute coronary syndrome, etc. (*n* = 192)	10 face-to-face CRs and 10 home CRs. Face-to-face CR; cycle ergometer; exercises Home CR; endurance training; strength training and stretching exercises; video conferencing 3 times a week.	Watts, METs, and wall square test: all improved significantly (*p* < 0.0001).	Videoconference software system
Ling Zhang (2022) [32]	Cohort study	COPD (*n* = 174, 46)	Home-based pulmonary tele-rehabilitation under telemedicine system. The follow-up time was 12 weeks. PR-1; 1~4 weeks, 5~20 times, 150~1200 min, *n* = 31; PR-2; 4~8 weeks, 20~40 times, 600~2400 min, *n* = 23; PR-3; 8~12 weeks, 40~60, 1200~3600 min, *n* = 40; PR-4; 12 weeks≦, 60 times≦, 3600 min≦, *n* = 34.	6mwt: cardiopulmonary capacity improved from 6.6 to 8.2 MET (*p* < 0.0001). Lower extremity muscle strength improved from 75.1 to 105.7 s (*p* < 0.0001). 6MWD significantly improved after 8 weeks.	Pulmonary tele-rehabilitation through the telemedicine system/monitoring HR and lung function
Monique Tabak (2014) [33]	Randomized controlled pilot trial	COPD (*n* = 14, 16)	Use of a tele-rehabilitation application with 3D accelerometer and smartphone at least 4 times/week for 4 weeks and regular rehabilitation.	Activity level (steps/day) showed no significant change over time. Health status improvement was non-significant between groups (*p* = 0.10) but significantly improved within the intervention group (*p* = 0.05). The activity coach was used 108% of the prescribed time.	3D accelerometer connected to smartphone via Bluetooth
Sang S Pak (2023) [9]	Single-center, parallel-group, randomized controlled trial	Patients with CSP (*n* = 41, 41)	Based on the results of the clinical assessment via video call, an 8-week tele-rehabilitation program was tailored to the participants’ needs. A device with motion digitization was used to quantify movements and transfer them to a mobile app. Real-time biofeedback was provided during exercise, along with video and audio cues to guide participants. Physical therapists monitored the data.	Both groups showed significant improvements in function as measured by the QuickDASH, with no significant differences between groups (–1.8, 95% CI –13.5 to 9.8; *p* = 0.75).	Inertial motion tracker with digitization of movements for biofeedback
Nurten Gizem Tore (2023) [34]	Randomized controlled trial	Patients diagnosed with moderate/mild KOA (*n* = 24, 24)	3 times/week, 45–60 min/session, for 8 weeks. Real-time exercises were performed via videoconference with a physical therapist.	Compared to the control group, the tele-rehabilitation group showed better scores on 30CST, IPAQ-SF, KOOS, QUIPA, treatment satisfaction, and EARS total scores and C subscales, as well as lower scores on NRS, HADS, TKS, and FSS. Statistically significant differences were found between the two groups in the B subscale of the EARS, but no significant differences were observed in the B subscale.	Zoom
Montse Nuevo (2022) [35]	Randomized controlled trial	Patients who underwent TKA (*n* = 23, 22)	For 4 weeks, a physical therapist used the ReHub^®^ to adjust the range and speed of movement and performed exercises accordingly. In addition, PT visited 2–3 times per week.	No significant differences were found between the groups in active or passive flexion range of motion (ROM) (*p* = 0.535; 0.680). However, both groups showed ROM gains during the 4-week period, with final ROMs exceeding 100°in both groups.	ReHub^®^, web platform, inertial motion sensor (IMU) incorporating accelerometer, gyroscope, and magnetometer
Villatoro-Luque FJ (2023) [36]	Single-blind, randomized controlled study	Nonspecific chronic low back pain (NCLBP) (*n* = 36, 35)	Received exercise instruction from a physical therapist and six exercises each week via instructional video on a computer platform.	Statistically significant time-by-group interactions were found in the strength of left hip flexors, right hip extensors with extended knee, and left hip extensors. Additionally, pain during hip flexion in the supine position was significantly different for both right (F = 5.133; *p* = 0.027) and left (F = 4.731; *p* = 0.033) hips.	
Gergüz Ç (2023) [37]	Randomized controlled trial	Junior tennis player (*n* = 20, 20)	2 days/week, 50 min/session, for 8 weeks; yoga and training with remote rehabilitation.	The yoga group showed significant improvements in core strength, stability, flexibility, balance (*p* < 0.001), and SF-36 scores for energy, mental well-being, social function, pain, and general health (*p* < 0.042).	Zoom

This table shows the effectiveness and safety of tele-physiotherapy for a variety of diseases using sensing technology.

## Data Availability

The original contributions presented in this study are included in the article/Appendix A. Further inquiries can be directed to the corresponding author.

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
