# Peer review of "Challenges and Prospects of Sensing Technology for the Promotion of Tele-Physiotherapy: A Narrative Review"

_sensors, 2024, doi:10.3390/s25010016_

Round 1
Reviewer 1 Report
Comments and Suggestions for Authors
As this is a review, I would expect this to be in the title. It is a narrative review, so perhaps that is the best term to use.
Page 1 Line 10 in the abstract, the word rapidly is repeated unnecessarily in the sentence.
I would like some rationale for why you limited your data base searches to Pubmed and Google Scholar. I would also expect you to report on the number of papers included in the study, some inclusion and exclusion criteria and the process you went through in deciding what to include from your searches.
I note there is a supplementary file that lists all the studies, but this detail should be reported briefly in the paper.
From lines 77-86 you make many statements that are informative and reasonable, however, you need to add citations from your included literature to support these statements in the results section.
At line 130 you refer to figure 1, but I cannot see a figure in your manuscript.
It would be helpful to have a table or a figure that outlines the results of your searches and how you narrowed in down to the included studies.
The table 1 is not well structure of formatted. You need to reduce the amount of text in the final column by summarising the findings. The current format is difficult for the reader to follow.
Line 136 you have left out a word.
The results section is not well structured. If you have done a literature review and identified a specific set of papers, that is what should be reported in an integrated way in the results. I am ot clear that you have limited your results section to the included literature from the way it is structured currently. I encourage you to synthesise the included literature and revise your headings to support this synthesis. The use of the conditions as headings does not achieve this synthesis and results in a lot of repetition and lack of cohesion in the reporting of the results.
In addition you have not demonstrated rigour in your approach to the searching and identification of articles. You need to describe this in more detail and include the overview of what was included and not included and why.
The discussion section is poorly structured and does not appear to be limited to the issues identified in the included articles, with additional items being added and new content raised. This is confusing for the reader. You need to limit your attention to the identified issues from your review articles and connect with relevant other articles from related areas to illustrate the points, not bring in totally new concepts here.
This paper requires a great deal of revision to bring it to a cohesive whole that will be helpful to the reader.
This is an important and timely topic and I encourage you to address the comments to bring this paper to a wider audience.
Comments on the Quality of English LanguageThe quality of the English language is generally good, but there are some word repetitions and missing words which require further editing.
Author Response
Thank you for the peer review. We have revised the report based on the comments we received. Our response to your comments is attached in WORD

Reviewer 2 Report
Comments and Suggestions for Authors
Summary
The work is a literature review of the last 5 years of work in motion sensing technology applied to telephysiotherapy
Appropriate sources are cited and the rationale for this work is well argumented: as sensors become more sophisticated, affordable and easier to integrate in our daily lives, they are being applied to remote phyisiotherapy, therefore research in the area is required.
Detailed Comments
There are some areas improvement which are mainly methodological. Some crucial details are not explained, or could be explained better:
How many papers were analysed in total, how many were excluded. Why pubmed and google scholar only? It is also unclear what query operators were used. Were the search terms used separately? Or were they all used using the OR operator. Consider including the queries.
It is not clear what criteria were used to prioritise works. For readers outside the field of physiotherapy it is difficult to understand if systems like VICON and Openpose are the most relevant systems at the moment. Please consider discussing whether these systems are the most prominent.
The number of articles retrieved (2205) is indicated for the queries to tele-rehabilitation and telehealth but not indicated for sensing technology articles.
Please reflect on ways to improve the readability and presentation of Table 1.
Did the author consider organising the "results" discussion by type of sensing technology? It could be useful to improve the structure and flow of the discussion.
Usability and Acceptability are (rightly) cited as important areas but the discussion does not mention which sources found this problem. Did all studies examined deal with usability and acceptability, and what factors influence acceptability by users?
Overall
Overall, the work does show that sensing technology used for tele physiotherapy is an active fields and there have been advancements in the period examined (2019-2024).
There are areas for improvement:
- Some crucial details on the search criteria could be included (e.g. queries used, n of articles examined, inclusion exclusion criteria). The fact that these details are missing impact the reproducibility of the study.
-The discussion on user acceptance and usability could have been expanded to include, for example what specific acceptance issues emerge in literature and if they emerge prominently or just in some of the studies.
Comments on the Quality of English LanguagePlease revise English for this work, the clarity of some sentences could be improved.
Please consider changing the word "sensitive" to "sensing" on line 23
There are instances of unnecessary hyphenation present throughout.
The use of the word "handling" on line 256 suggests that there are ways in which humans (researchers, medical professionals, patients) deal with the acquired data, however the discussion which follows deals with a variety of issues which are not necessarily included in the word "handle"/
Author Response

(The authors gave the same response as above.)

Round 2
Reviewer 2 Report
Comments and Suggestions for Authors
All suggestions and comments made by this reviewer were addressed. English was also improved.